# Increased Collagen Turnover Impairs Tendon Microstructure and Stability in Integrin α2β1-Deficient Mice

**DOI:** 10.3390/ijms21082835

**Published:** 2020-04-18

**Authors:** Daniel Kronenberg, Philipp A. Michel, Eva Hochstrat, Ma Wei, Jürgen Brinckmann, Marcus Müller, Andre Frank, Uwe Hansen, Beate Eckes, Richard Stange

**Affiliations:** 1Department of Regenerative Musculoskeletal Medicine, Institute for Musculoskeletal Medicine, Westfälische Wilhelms-University, 48149 Münster, Germany; Daniel.kronenberg@ukmuenster.de (D.K.); E.hochstrat@uni-muenster.de (E.H.); mars-ma@163.com (M.W.); 2Department of Trauma-, Hand-, and Reconstructive Surgery, University Hospital Münster, 48149 Münster, Germany; Philipp.michel@ukmuenster.de; 3Institute of Virology and Cell Biology & Department of Dermatology, University of Lübeck, 23562 Lübeck, Germany; brinckmann@vuz.uni-luebeck.de; 4Department of Molecular Medicine, Institute for Musculoskeletal Medicine, Westfälische Wilhelms-University, 48149 Münster, Germany; marcus.mueller@ukmuenster.de (M.M.); andre.frank@ukmuenster.de (A.F.); uhansen@uni-muenster.de (U.H.); 5Translational Matrix Biology, University of Cologne Medical Faculty, 50931 Koln, Germany; beate.eckes@uni-koeln.de

**Keywords:** tendon biology, collagen, integrin α2β1

## Abstract

Integrins are a family of transmembrane proteins, involved in substrate recognition and cell adhesion in cross-talk with the extra cellular matrix. In this study, we investigated the influence of integrin α2β1 on tendons, another collagen type I-rich tissue of the musculoskeletal system. Morphological, as well as functional, parameters were analyzed in vivo and in vitro, comparing wild-type against integrin α2β1 deficiency. Tenocytes lacking integrin α2β1 produced more collagen in vitro, which is similar to the situation in osseous tissue. Fibril morphology and biomechanical strength proved to be altered, as integrin α2β1 deficiency led to significantly smaller fibrils as well as changes in dynamic E-modulus in vivo. This discrepancy can be explained by a higher collagen turnover: integrin α2β1-deficient cells produced more matrix, and tendons contained more residual C-terminal fragments of type I collagen, as well as an increased matrix metalloproteinase-2 activity. A greatly decreased percentage of non-collagenous proteins may be the cause of changes in fibril diameter regulation and increased the proteolytic degradation of collagen in the integrin-deficient tendons. The results reveal a significant impact of integrin α2β1 on collagen modifications in tendons. Its role in tendon pathologies, like chronic degradation, will be the subject of future investigations.

## 1. Introduction

Integrins are a family of heterodimeric type I transmembrane receptors. Eighteen α- and 8 β-subunits have been characterized so far, leading to 24 described combinations in humans [1]. Having a short cytoplasmic region, which interacts with the cytoskeleton [2,3], and a large extracellular part, integrins play central roles in cell attachment, as well as in cell-matrix signaling [2,4,5,6]. A group of integrin α-subunits, including the collagen-binding receptors, contains another structural feature, an additional domain called the αA-domain [3]. All these α-subunits pair with the β1 integrin subunit and are able to bind to collagen triple-helices [7,8]. They can be blocked by the collagen-mimicking peptide sequence GFOGER [9,10]. Members of this group are integrin α1β1, α2β1, α10β1 and α11β1. α1β1 and α10β1 integrin have a comparable lower affinity to the major fibrillary collagens and bind to collagens of the basement membrane, like collagen type IV [11]. Integrin α2β1 and α10β1, which bind to collagen type II in cartilage, do not bind the collagen fibrils directly but interact with collagen-associated macromolecules. Isolated, accessible collagen molecules still serve as a valid binding partner for integrin α2β1, α10β1 and α11β1, leading to the conclusion that integrin binding detects collagen molecules in the damaged extra-cellular matrix and serves as a sensor for tissue damage. This holds true for collagen type II and chondrocytes. For collagen type I, the situation may be indeed completely different, as the putative binding site for collagen-binding integrins, which is deemed to be cryptic, was shown to be accessible due to a proline-mediated side-specific flexibility discovered by structural analyses. [12,13] Thus, integrin α2β1 might play a more complex role in the tissue homeostasis of collagen type I-rich tissue, like bones, skin and tendons.

While most information on the function of integrin α2β1 has been derived from studies of platelets and vasculature, this integrin is also expressed by fibroblasts and other mesenchymal cell types [14]. As an adhesion receptor, integrin α2β1 plays major roles in cell-matrix communication in tissue repair [15,16] and cancer [17]. Inhibiting integrin α2β1 with blocking antibodies results in decreased bone morphogenic protein (BMP)—mediated mineralization that crucially depends on the time of application during cell differentiation [18]. Using the constitutive integrin α2β1 knockout mouse, we demonstrated in previous studies that its absence in bone cells led to increased production of collagen type I in vitro, as well as in situ, which mitigated the impact of age-related bone degradation in aging mice. On a molecular level, we could show that the overproduction of collagen led to an altered morphology in collagen fibrils [19]. Bone collagen fibrils usually have diameters between 35–50 nm, depending on gender, age and species [20]. In integrin α2β1-deficient animals, however, the fibrils appeared amorphous and fused with an increased diameter. Our results pointed to a regulatory function of integrin α2β1 in collagen morphology and homeostasis; therefore, we focused in this study on tendons, another collagen-based musculoskeletal tissue that severely relies biomechanically on a strictly ordered arrangement of collagen fibrils. 

Most of the dry mass of the tendon consists of collagen fibrils, mainly collagen type I. Besides, tendon tissue shows low cellularity of primarily tenocyte-like fibroblasts. Discrimination between fibroblasts and tenocytes appears to be difficult, since both have a spindle-shaped morphology. Tenocytes are even more elongated than other fibroblasts and are positive for the markers tenomodulin and scleraxis, although fibroblasts can fit these features as well [21]. Collagen fibrils are associated in a hierarchical structure, initially forming fibrils that form primary, secondary and tertiary fiber bundles, which eventually form the tendons [22]. Collagen fibrils are surrounded by extrafibrillar matrix components which include proteoglycans like aggrecan, biglycan, decorin, fibromodulin, lumican and versican [23]. Collagen fibrils are important for the tensile strength and loading capacity of tendons, whereas proteoglycans can bind large quantities of water, providing protection against compressive forces and contributing to the overall flexibility of the tissue. The collagen-associated proteoglycans such as biglycan and decorin are also involved in the determination of the collagen fiber thickness [24]. Taken together, collagen biology plays an important role in the morphological, as well as functional, characteristics of the tendon. Therefore, influencing variables are of great interest. 

The aim of this study was to investigate the impact of α2β1 integrin on the structural and functional properties of the tendon. 

Bayer et al. already showed that integrin α2β1 plays an implied role in tendon mechano-sensitivity, since they observe an increased protein presence when their utilized tendon construct was cultivated under a de-tension condition [25]. The aim of this study was to investigate the impact of α2β1 integrin on the structural and functional properties of the tendon. In the case of, we hypothesize that communication between cells and the extracellular matrix would be impaired in the absence of integrin α2β1, leading to compromised tendon function and morphology. 

## 2. Results

### 2.1. Integrin α2β1-Deficient Tenocytes Produce More Collagen and Express More Matrix and Tendon-Related Proteins In Vitro

Investigating a protein involved into cell attachment and proliferation, the morphology of tenocytes was accessed prior to the biological and biochemical comparison between integrin α2β1-deficient tenocytes and the wild-type. In this context, the genotypes did not show any apparent differences (Figure 1A). Isolated tenocytes cultivated for three weeks produced a sufficient matrix for quantification by the total collagen assay detecting hydroxyprolines. Since type I and type III collagen are the major contributors of the organic matrix in tendons, these findings reflect the overall collagen production. In line with isolated integrin α2β1-deficient osteoblasts, cultivated tenocytes lacking integrin α2β1 also produced more collagen (Figure 1B). This increase was controlled at the transcriptional level, as quantitative real-time PCR revealed increased (by 100%) expression of collagen type I and type III. Interestingly, we also detected increased expression of scleraxis, which is a marker for tendons, especially at an embryonic state (Figure 1C). 

To rule out that all the effects caused by integrin α2 deficiency are caused by an increased compensator expression of the other collagen-binding integrins, a qRT-PCR with tenocytes was performed (Appendix A). For integrin α1 and α11, the expression was quite similar when comparing wild-type and integrin α2β1-deficient tenocytes. Integrin α10 showed a slightly increased expression level but with a high standard deviation. Additionally, the overall expression levels of integrin α10 were very low: one-quarter of the integrin α11 and one-half of the integrin α10 expression (data not shown). There was no compensation of the other collagen-binding integrins found, which is similar to bone cells.

### 2.2. Increased Collagen Expression by Integrin α2β1-Deficient Tenocyte Like Fibroblasts Is Not Transferred to the Tissue

When measuring the overall collagen content by creating HCl-hydrolysates of whole tendons, integrin α2β1-deficient tendons did not show increased levels (wild-type 19.28 mg collagen per g tendon +/- 11.11 and integrin α2β1^-/-^ 17.14 mg collagen per g tendon +/- 9.531, N = 7; Figure 1D). To rule out that the integrin-deficient tendons contained reduced cell numbers causing collagen quantities as well, we counted the tendon-resident cells in four 1-µm² grids per tendon section (Figure 1E). As the cell count per mm² did not show statistically significant differences (wild-type: 1093+/−339.3 and integrin-deficient tendons: 1130 +/−261, N = 12; Figure 1F), we concluded that augmented collagen production in cell culture was somehow mitigated in the tissue. 

### 2.3. Collagen Fibrils Lacking Integrin α2β1 Have No Increased Inherent Disorder

Since collagen fibrils in the bone of integrin α2β1-deficient mice showed a phenotype in which no uniform fibril diameter could be observed and fibrils were fused, the investigation of fibril appearance was also part of this study. Polarization microscopy utilizes the quasi-crystalline state of collagen fibrils. If collagen fibrils are in disorder, the light passes directly through the specimen, but when fibrils are arranged in proper order, they are interfering with the polarized light. This can be quantified by the average gray value of the obtained image, as shown in Appendix A. Integrin α2β1-deficient tendons did not yield significantly abnormal values, reflecting an overall normal organization. 

### 2.4. Integrin α2β1-Deficient Tendons Show a Predominance of Small Collagen Fibrils

The collagen fibrils in the Achilles tendon are among the widest found in an organism. The mean diameter of the collagen fibrils in the murine Achilles tendons was 206.44 nm. In the integrin α2β1-deficient background, more fibrils with a thinner diameter were visible. Quantifying the mean diameter in these tendons, a significantly reduced diameter of 144.28 nm was measured. With the evaluation of the distribution of the fibril diameters, two distinct populations at ~100 nm and at 220–250 nm became obvious (Figure 2). The integrin α2β1-deficient animals have more and thinner fibrils when compared to the wild-type tendons. 

### 2.5. Biomechanical Comparison Reveals Reduced Young’s Modulus in the Absence of Integrin α2β1

To see if the altered collagen fibril morphology in integrin-deficient tendons would affect the biomechanical properties of these tendons, we subjected them to pulling tests. In line with smaller diameters, the cross-sectional area was also reduced in the integrin-deficient tendons. Accordingly, the load to failure was also significantly reduced in those tendons (Appendix A). The static E-modulus tended to be reduced in the integrin-deficient tendons, although this was not significant. The analysis of the dynamic E-modulus showed no differences at a 4% strain. With increasing strain rates, the integrin group showed a less pronounced rise of the dynamic E-modulus compared to the wild-types. This effect was greatest at an 8% strain (Figure 3).

### 2.6. Integrin α2β1-Deficient Tendons Showed Decreased Lysyloxidase Quantities—The Cross-Linking Pattern Appeared Not to Be Affected

The dynamic biomechanical tests demonstrated a clear impairment of tendon stability at higher strains and raised the question if the thinner collagen fibrils and reduced tendon stability might be due to overall changed collagen morphology or alterations in the cross-links. One major protein involved in forming cross-links is lysyl oxidase. This enzyme catalyzes the oxidative deamination of lysine or hydroxylysine residues located in the telopeptide region of the collagen molecule, leading to aldehydes. Subsequently, condensation of the aldehydes with juxtaposed amino acid residues in adjacent collagen chains results in the formation of cross-links. It showed a distinct band at ~40 kDa (Appendix A). After quantification and setting the signal for wild-type LOX/GAPDH as 100%, the integrin α2β1-deficient tendons yielded a signal of 57% (Figure 4A,B). Quantification of the signals revealed a reduction by 43% of LOX amounts in integrin α2β1-deficient tendons compared to controls.

Next, we investigated if reduced lysyl oxidase amounts would translate to reduced lysyl oxidase-mediated cross-links in integrin α2β1 tendons. Cross-link analysis showed a slight but significant reduction of the difunctional cross-link hydroxylysinonorleucine (HLNL) for integrin α2β1-deficient tendon (Figure 4C). By contrast, the difunctional cross-link dihydroxylysinonorleucine (DHLNL) showed a no-change status (Figure 4D). In addition, the mature cross-link HP and HHMD were unchanged (data not shown). However, as a byproduct of the amino acid analysis, we found that the integrin α2β1 tendon displayed a striking loss of non-collagenous proteins, resulting in a decrease of the non-collagenous protein:total protein ratio in the integrin α2β1-deficient tendon (Figure 4E). The wild-type tendon was made up out of 36.8% non-collagenous proteins, whereas the integrin α2β1-deficient tendon contained 23.7% of these proteins (*p* = < 0.0001).

### 2.7. Integrin α2β1-Deficient Tendons Show Increased Gelatinase Activity

The presence of thinner fibrils and absence of non-collagenous proteins and similar collagen content by functional increased collagen expression of the tenocytes raises the question about the fate of the proteins. An indicator of increased turnover and remodeling is thus an increased proteolytic activity. Therefore, gelatin zymography of tendons was carried out that reveals the activity of MMP-2 and MMP-9. Results showed a prominent band at ~69 kDa and a weaker one at ~60 kDa. These bands correspond to the pro and mature forms of MMP-2. Quantification of the proteolytic zone corresponding to MMP-2 activity revealed a significantly larger area (four-fold compared to wild-type) in the integrin α2β1-deficient tendons (Figure 5a,b). This increased proteolytic activity points to an increased collagen turnover in integrin α2β1-deficient tendons. We also assayed the activity of non-gelatinase MMPs; however, casein zymography that reflects the activity of MMP-3, -8 and -13 did not reveal proteolytic zones (data not shown), which excludes these MMPs as contributors to increased collagen turnover in integrin mutant tendons.

### 2.8. Integrin α2β1-Deficient Tendons Contain More Soluble Collagen Fragments

To substantiate our observation of increased collagen turnover further, we determined the amounts of soluble collagen fragments, which are an indicator for proteolytic activity, since uncleaved collagen remains incorporated in the fibrils. Quantification of the soluble collagen C-terminal fragments (CTX) by ELISA indeed confirmed higher levels in integrin-deficient tendons: We determined a mean concentration for the CTX fragments of 2.583 ng per mg total protein in wild-type and 3.659 ng per mg total protein in integrin α2β1-deficient tendons (Figure 5c). This result concords with an increased MMP-mediated turnover and indicates that overall collagen remodeling in integrin α2β1-deficient tendons is enhanced in comparison to the wild-type. 

## 3. Discussion

In the present study, we investigated the influence of integrin α2β1 on collagen metabolism in tendon tissues. We demonstrated at a cellular level that integrin α2β1-deficient tenocyte-like cells expressed higher quantities of collagen, as well as scleraxis, which led to an increased quantity of newly formed extracellular matrix in cell cultures. This was in line with our findings characterizing the absence of integrin α2β1 in bones [19]. Expression and production of collagen I by bone-forming osteoblasts, as well as the collagen content in knockout bones, was increased as well. Additionally, EM images of α2β1-deficient bones showed a highly dysregulated organization with collagen fibrils, which had fused to amorphous structures [19]. All these findings point to the conclusion that integrin α2β1-deficient tissues might be more prone to fibrosis. However, the main characteristics of fibrosis were not met in the tendons of our study [26]. The major characteristics of fibrosis are the irregularity of collagen fibrils and the increased deposition of collagen fibrils, leading to scar-like tissue. We could show that the microstructural morphology of the integrin α2β1-deficient tendons exhibited remarkable differences to the collagen network in bones. Even though the integrin α2β1-deficient tenocyte-like cells expressed and produced more collagen, the overall collagen content in tendons was not increased, and the cell-to-matrix ratio was not altered. The intrinsic disorder of the tendons on the light-microscopy level was also not affected, and electron microscopy revealed that the collagen fibrils in mutant tendons were distinct and evenly formed. However, when compared to wild-type tendons, mutant fibrils had an overall decreased diameter, which were composed of a population of thin fibrils and a population of wild-type-like fibrils of thicker diameters. Of note, these structural alterations translated into a compromised stability and reduced biomechanical function of mutant tendons. In line with the morphological finding of altered fibril distributions were the decreased cross-sectional area and the reduced load to failure in mutants compared to the wild-type tendons. Under rising physiological strain, the tendons presented a less pronounced increase in dynamic E-modulus. This attests that the biomechanical alterations were influenced by the difference in fibril populations. 

Reduction of fibril thickness alone does not lead to decreased biomechanical weakness, which has been shown by Shu et al., who demonstrated that tendons of perlecan [27]-deficient animals had increased tensile stability. In their investigations, the knockout of perlecan caused an overall decreased collagen fibril diameter similar to the situation in our integrin α2β1-deficient tendons. They explained the superior biomechanical properties with an altered collagen organization. However, in contrast to our findings, they got a uniform change and decrease in the collagen fibril diameter, whereas we could also detect a portion of mature fibrils >200 nm in the integrin α2β1-deficient tendons. These singular thicker fibrils may disrupt the quasi-crystalline structure, introducing a different kind of disorder without disturbing the fiber orientation. Together with the constant remodeling, a disruption of tight packing would lead to decreased biomechanical properties.

Investigations of further sources of decreased tendon qualities focused on cross-links. We identified a reduced presence of lysyl oxidase in integrin α2β1-deficient tendon tissues. This is in line with further findings linking the expression of lysyl oxidase and integrin α2β1, which has been demonstrated in heart tissues [28]. However, our analysis of the cross-links did not support a decreased lysyl oxidase activity but an overall lack of non-collagenous proteins like proteoglycans. 

In the absence of integrin α2β1, we could quantify an overall reduction of all non-collagenous proteins. This may explain biomechanical and morphometric differences between the two knockout models. The lack of proteoglycans was also investigated by Dourte et al., who characterized the effects of small leucine-rich proteoglycans (SLRPs), such as biglycan, on tendon biomechanics using the biomechanical methods adapted in this study. They were observing an increased biomechanical stability when biglycan was missing. However, the effects on other non-collagenous proteins were not described. It might be the case that other SLRPs like decorin can compensate for this phenotype, as Dunkman et al. showed on the expression level in [29]. 

Integrin-deficient tenocytes express and produce more collagen, but this does not appear in the overall tendon tissue. Consequently, we suspected that proteolytic degradation would cause an increased collagen turnover. This was confirmed by increased levels of MMP-2 activity, which is a potential candidate for collagen degradation in tendons. It has already been shown that this proteinase, which is able to cleave collagen type I under certain conditions, was upregulated in the tendon tissue when it was physically challenged [30,31]. Since we showed that integrin α2β1-deficient tendons were more sensitive to non-physiological stress (strain rates >6%), this might explain the increased MMP-2 activity in our model. The increased quantity of soluble C-terminal collagen fragments, which are products of proteolysis, supported this finding. This biomechanical stress was missing in the tendon cell cultures; therefore, we observed just the overexpression of collagen without the expression of collagen-degrading enzymes.

Integrin α2β1 deficiency revealed that just the plain production and apposition of collagen is not sufficient to explain solely peak biomechanical properties of tendon tissues. The size of the fibrils, as well as the conformity within the tendon tissue, are of utter importance. This is achieved by enzymes like lysyl oxidase and proteoglycans, which both seem to be impaired by the lack of integrin α2β1.

## 4. Materials and Methods 

### 4.1. Animals Used 

Mice lacking the integrin α2 subunit were generated by Holtkoetter et al. in a C57BL/6 background [32]. Genotypes were confirmed by PCR, as described previously [19]. The mice used in this study were female, and littermates were genotyped as wild-type (wild-type, wt) or homozygous mutant (*Itga2*^−/−^). Animals were euthanized as young adults (12–16 weeks) by cervical dislocation. Achilles tendons were harvested from hind legs. All procedures involving mice were approved by the LANUV (North-Rhine Westphalia, Germany) under the reference 84-02.05.50.15.005.

### 4.2. Histology

Achilles tendons dissected from right hind limbs of mutant and control mice at 16 weeks of age were processed for histological examination. Eight tendons per group were harvested and dissected from the bone insertion site at the calcaneus. Surrounding soft tissue was carefully removed. The tendons were fixed in paraformaldehyde 4% overnight at 4 °C in the dark, decalcified in ethylene diamine tetra acetic acid (EDTA) for 2 weeks and dehydrated in a digressive series of ethanol. After embedding in paraffin, longitudinal sections (5 μm thickness) were cut and stained with picrosirius red 0.1 g/100 mL (Direct Red 80) for evaluation using polarized light. Hematoxylin and eosin (HE) staining was used for general histology and assessing the total cell number. An image analyzing system (Olympus BX51 microscope, Olympus, Hamburg, Germany) with the Cell Sense software (Olympus SIS, Münster, Germany) with an additional polarizing filter was used to obtain the cell number and average gray value.

### 4.3. Polarization Microscopy

Tendon sections were examined under monochromatic polarized light. The long axis of the tendon was aligned in 45° to the polarized light. Images were taken under identical illumination and analyzed using Image J (https://www.ncbi.nlm.nih.gov/pubmed/?term=22930834). Birefringence measured as brightness was assessed measuring the average gray value (1–255), where high gray values correlated with superior collagen fiber organization [24]. Therefore, gray-scale images were generated, and the average gray value was measured in five randomly selected regions of interest (ROI) per tendon.

### 4.4. Transmission Electron Microscopy

The tendons were fixed overnight with 2% formaldehyde and 2.5% glutaraldehyde dissolved in 100-mM cacodylate buffer (pH 7.4). After washing with phosphate-buffered saline, the samples were post-fixed in 0.5% osmium tetroxide with 1% potassium hexacyanoferrate (III) in 0.1-M cacodylate buffer for 2 h. After washing the samples with water, they were dehydrated in a descending ethanol series. Afterwards, the samples were incubated in propylene oxide two times and embedded in Epon. Ultra-thin transversal sections were captured on grids and contrasted with 2% uranyl acetate. Images were acquired with a Philips EM-410 electron microscope (EM) at 60 kV. For integrin α2^-/-^ and wild-type animals, four different tendons each were analyzed, and the diameters of 1600 collagen fibrils per phenotype were determined using Image J (https://www.ncbi.nlm.nih.gov/pubmed/?term=22930834). 

### 4.5. Biomechanical Testing

The samples were mounted in a custom-built test set-up consisting of a metal pin under which the calcaneus was attached in a physiological angle of 45° dorsal extension and a plastic clamp in which the proximal end of the tendon was clamped with the aid of sandpaper, as described in [33]. The distance between both clamps was set to 5 mm. Afterwards, the fastened tendon was transferred into a phosphate-buffered solution (PBS) bath at room temperature. Two cameras (DigiMicro Profi, (dnt, Dietzenbach, Germany) and Dino-lite Edge 3.0 (Dino-lite, Almere, The Netherlands)) and the corresponding software were used to determine the dimensions of the tendons from two angles. Three measurements of the midsections of the tendons were registered and averaged to obtain the cross-sectional area of the tendons assuming an ellipsoid form of this area. According to the measured cross-sectional area, tendons were stretched load-adjusted to an individually calculated preload of 0.1 MPa. Changes in length were monitored until no further elongation of the tendon was measurable. Then, initial length of the tendon was measured. Tendon biomechanics can best be explained with a viscoelastic model; therefore, we used a testing protocol, which we adapted from Dourte et al. and which was previously used to analyze the dynamic biomechanical features of tendon tissues [33]. This protocol includes a preconditioning phase with a cyclic loading between 0.5% and 1.5% strain at 0.25 Hz. Samples were stretched once to 4% to simulate physiological strain. For measurement, a 10-min stress-relaxation test was performed, followed by subsequent cyclic loading in a frequency sweep (0.1 Hz, 1 Hz and 5 Hz) with an amplitude of 0.125%. Analogous tests were conducted at strain levels of 6% and 8%. Finally, tendons were loaded until failure in a displacement-controlled load-to-failure-ramp with a speed of 0.1% strain per second. The mode of failure was registered. The data analysis was performed with a customized MATLAB script (R2017a). Dynamic Young’s modulus was calculated from the amplitude ratio of the stress-time-curve/strain-time-curve. It represents the tissue’s resistance against deflection. Static Young’s modulus was calculated from the linear-elastic region of the load-to-failure-curve.

### 4.6. Tenocyte Culture

Murine Achilles tendons were dissected free of surrounding soft tissue, and the tendons were washed in PBS. They were divided into 1-mm pieces and transferred into a tube (15 mL) containing a digesting medium (0.3 % collagenase I in αMEM high glucose). After 1.5 h shaking at 37 °C, the cell suspension was centrifuged at 2000 rpm for 10 min. The cells cultured using αMEM with added 10% FCS (Invitrogen, Carlsbad, CA, USA), 1 mM L-ascorbic acid 2-phosphate, 100 u/mL penicillin and streptomycin. Medium was replenished when the primary tenocytes had attached to the culture dish. Afterwards, medium was changed every two days until cells reach 80%–90% confluence. Cells were used in passages 1–3. 

### 4.7. Quantitative Real-Time PCR

Tenocytes were lysed using the RLT buffer provided with the RNeasy Micro Kit (Qiagen, Hilden, Germany) which was used as well for the following RNA isolation. RNA was transcribed into cDNA by reverse transcriptase (RT) polymerase chain reaction (Omniscript RT Kit, Qiagen, Hilden, Germany) using oligo-dT-Primer (Operon, Tokyo, Japan). The primer sequences (Metabion GmbH, Planegg, Germany) used for quantitative real-time PCR (qRT-PCR) are shown in Table 1 QRT-PCR was performed using the DyNAmo Flash SYBR Green qPCR Kit (Biozym, HessischOldendorf, Germany) in a Bio-Rad IQ5 thermocycler with the corresponding IQ2 detection module (Munich, Germany). Each experiment included a negative control using water as the template. Expression of each gene was normalized to the expression of hypoxanthine guanine phosphoribosyl transferase (HPRT, Accession No. NT039702).

### 4.8. Collagen Content in Tenocyte Generated Matrix and in Tendon Tissue

For collagen determination, tenocytes were kept in culture for 3 weeks. The soluble fraction was washed off with PBS, and the tenocytes with their generated matrix were resuspended in 12-M HCl. For tendon tissue, the tendon was carefully removed from its sheath, and a small piece of the central area was taken. The tendon pieces were dissolved in 12-M HCl at a concentration of 100 mg tendon per ml. The concentration of hydroxyproline-reflecting collagen was determined using a Total Collagen Assay Kit (QuickZyme, Leiden, The Netherlands) with rat-tail tendon collagen type I as standard.

### 4.9. Gelatin-/Casein Zymography of Tendons

Tendons of both genotypes were dissected as described and stored in liquid nitrogen until usage. The tendons were disrupted using the Precellys Steel Kit 2.8 mm together with the Precellys 24 Homogenizer (Bertin, Frankfurt am Main, Germany). Three-hundred microliter lysis buffer (50-mM Tris-HCl, pH 8, in 2% SDS) was added to each Achilles tendon. The tissue was ground at 5000 rpm in three cycles, each 60 s. Each cycle was followed by three minutes of cooling down at room temperature. The tissue homogenate was centrifuged to remove foaming. Sample of 30 μg was separated under nonreducing conditions on a polyacrylamide gel electrophoresis with 0.1 % gelatin (bovine, Sigma Aldrich, Taufkirchen, Germany) and 12% acrylamide. After the run, the gel was washed 2× for 30 min in 2.5% Triton X-100. Then, the detergents were removed by washing 4× for 5 min in demineralized water. To activate the resident proteinase, the gel was incubated for 16 h in 50-mM Tris-HCl, pH 8.5 + 5-mM CaCl2. The gel was stained thereafter with Coomassie brilliant blue, and de-stained areas were quantified using the Image Studio software (Li-Cor, Bad Homburg, Germany).

### 4.10. Lysyl Oxidase Determination

Tendons were lysed as described above. Prior to Western blotting, 30 µg of the tendon lysate was separated using SDS-PAGE (10% acrylamide). The proteins were transferred onto immobilon FL membranes using a semi-dry blotting technique. Lysyl oxidase was detected with the LOX antibody (E-19, sc-32410, Santa Cruz Biotech, Heidelberg, Germany, 1:1000), followed by incubation with 800 CW donkey anti-goat secondary antibody (Li-Cor, Bad Homburg, Germany, 1:20.000). A Li-Cor Odyssey CLx was used as the detector. The signal was normalized to GAPDH (Proteintech, Manchester, UK, 1:1000) using a 680 RD goat anti mouse secondary antibody (Li-Cor, Bad Homburg, 1:20.000) for detection. Bands were quantified using the image studio software provided with the Odyssey CLx. The mean of the lysol oxidase/GAPDH signal of the wild-type samples were set to 100%.

### 4.11. Assessment of Collagen, Non-Collagenous Protein and Collagen Cross-Links

Collagen and non-collagenous protein levels, as well as the abundance of the collagen cross-links dihydroxylysinonorleucine (DHLNL), hydroxylysylpiridinoline (HP), hydroxylysinonorleucine (HLNL) and histidinohydroxymerodesmosine (HHMD), were determined in tendons as described previously for adult mouse lungs and human tissues [34,35]. The nomenclature used in this study refers to the reduced variants of the intermediate collagen cross-links (DHLNL, HLNL and HHMD). The content of collagen and of non-collagenous proteins was analyzed on an amino acid analyzer in the supernatant and the insoluble residuum of a sample after collagenase digestion. 

### 4.12. Quantification of Soluble Collagen Fragments in Tendons

For quantification of the soluble collagen fragments, tendons were lysed as described in the zymography paragraph. Insoluble, intact collagen fibrils were removed from the lysate by centrifugation for 10 min at 13.200× *g*. The samples were diluted 1:10, and a CTX ELISA detecting the C-terminal telopeptides of type I collagen was performed as instructed by the manufacturer (Wuhan USCN Business Co., Ltd., Houston, TX, USA).

### 4.13. Statistical Analysis

Statistical analysis of the data was carried out using GraphPad Prism software (Graph Pad Software Inc. GraphPad Software, San Diego, CA, USA). Mann–Whitney U test was used to determine significance of the data. *P*-values were classified as follows: * *p* < 0.05, ** *p* < 0.01, *** *p* < 0.001 and **** *p* < 0.0001. Two-way ANOVA was used for the dynamic biomechanical testing at different strain rates.

## 5. Conclusions

The present study revealed an accelerated tendon collagen turnover in the absence of integrin α2β1. Albeit having a comparable collagen content to wild-type, integrin α2β1-deficient tendons contain higher quantities of residual collagen C-terminal fragments; we could also demonstrate that isolated tenocytes in vitro were capable of producing more collagen, similar to integrin α2β1-deficient osteoblasts. With the increased MMP-2 activity, we identified a potential candidate for degradation, pointing to insufficient matured collagen fibrils present in the integrin α2β1-deficient tendons. This would explain the higher quantity of thinner fibrils, leading to impaired biomechanical properties. Furthermore, the absence of non-collagenous proteins also seemed to contribute to the biomechanical weakening of the tendons. Future investigations will elucidate whether integrin α2β1 may play a relevant role in spontaneous human tendon ruptures due to degeneration.

## Figures and Tables

**Figure 1 ijms-21-02835-f001:**
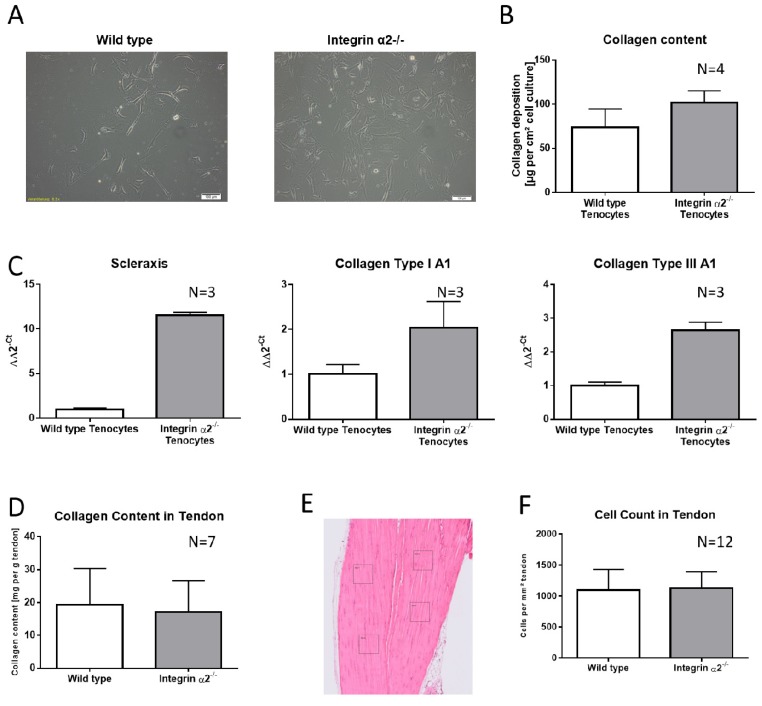
Collagen content differs in isolated cells compared to tendon tissue. Tenocyte-like cells isolated from the Achilles tendon of wild-type and integrin α2β1-deficient mice (**A**). Total collagen content of the matrix produced by tenocytes after 3 weeks of culture (**B**). qPCR analysis of key genes in isolated tenocytes (**C**). Total collagen quantification of wild-type and integrin α2β1-deficient Achilles tendons from 4-month-old female mice (**D**). HE staining of a representative longitudinal slice of an Achilles tendon with the cell quantification pattern (**E**). Cell quantity in situ (**F**).

**Figure 2 ijms-21-02835-f002:**
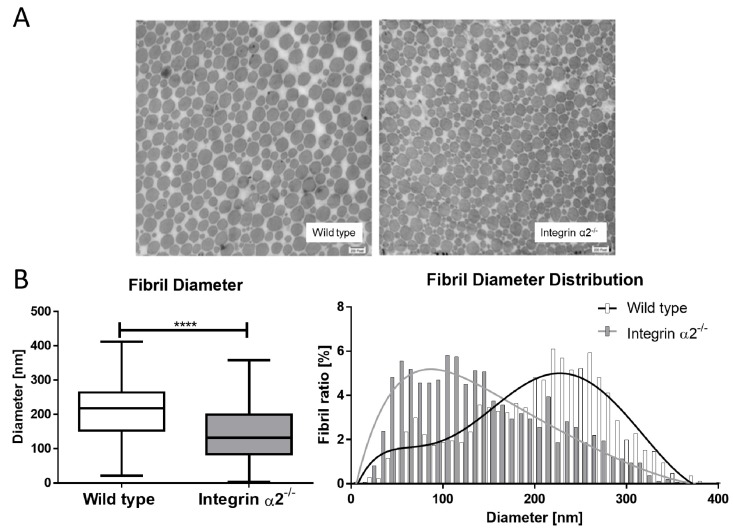
Collagen fibrils are thinner in integrin α2β1-deficient tendons. Overview of a transmission electron micrograph of integrin α2β1-deficient and wild-type tendons (**A**). Distribution of fibril ration to diameter, and >1600 fibrils/genotype from four different tendons were measured (**B**). Mann-Whitney U test was used for statistical testing (**** *p* < 0.0001).

**Figure 3 ijms-21-02835-f003:**
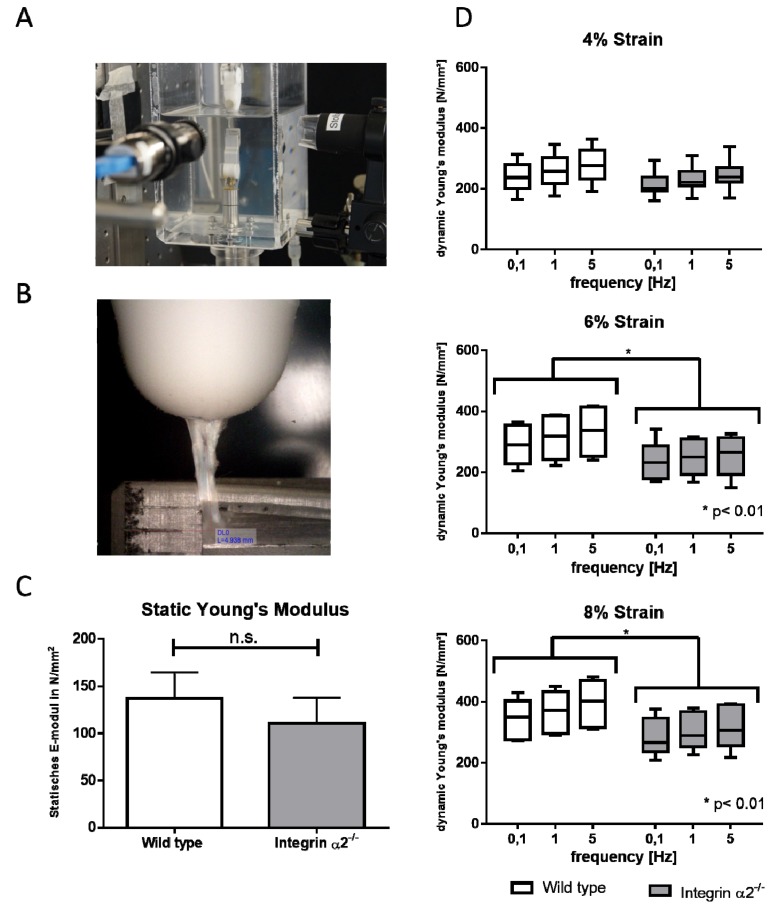
Biomechanical testing set-up (**A**) with the mounted tendon (**B**). Static Young’s modulus at the load to failure (**C**). Dynamic Young’s modulus measured at 0.1 Hz, 1 Hz and 5 Hz straining the tendon at 4%–8% (**D**) N = 8. Statistic analysis: Mann-Whitney U test was used for the static biomechanical testing (n.s. *p* > 0.05), and two-way ANOVA was used for the dynamic biomechanical testing, (* *p* < 0.01).

**Figure 4 ijms-21-02835-f004:**
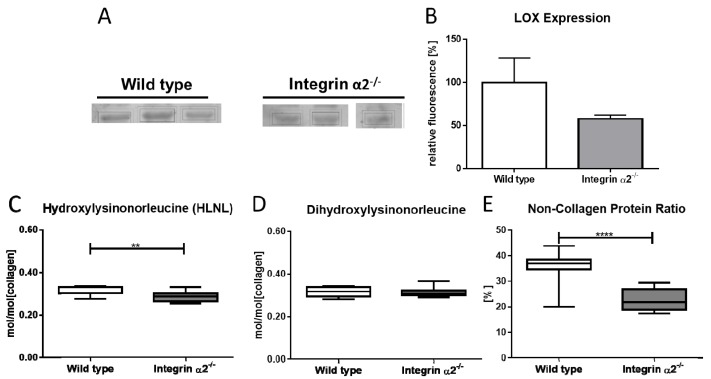
Lysyl oxidase and its activity in integrin α2β1-deficient tendons (**A**). Ten percent polyacrylamide gel electrophoresis with consecutive Western blot detecting for lysyl oxidase. Detection was done in a Li-Cor Odyssey CLx fluorescence scanner using a fluorescent secondary antibody. Quantification was done using the provided image studio software (N = 3, **B**). Cross-link analysis of integrin α2β1-deficient tendons (N = 12) showed a slight decrease for hydroxylysinonorleucine (HLNL, C) and a no-change status for dihydroxylysinonorleucine (DHLNL). (**D**). Amino acid analysis revealed a marked decrease of the ratio of non-collagenous protein:total protein (**E**). Mann-Whitney U test was used for statistical testing (** *p* < 0.01, **** *p* < 0.0001).

**Figure 5 ijms-21-02835-f005:**
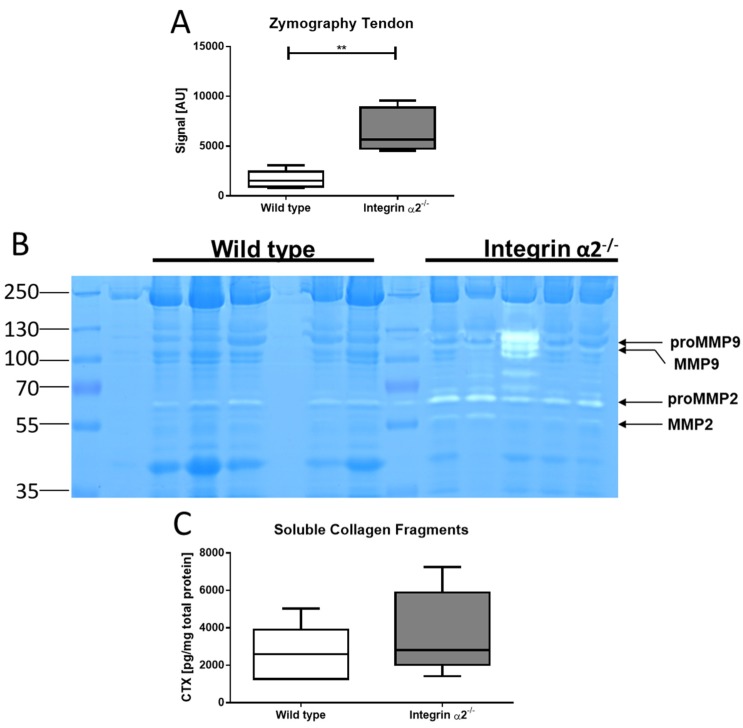
Resident MMP-2 activity is increased in integrin α2β1-deficient tendons. Gelatin zymography of 30-µg tendon lysate. Quantification of the MMP-2-activity signal (**A**). Zymography gel 10% acrylamide with polymerized gelatin. The arrows depict the pro and active forms of MMP-9 and MMP-2, respectively (**B**). C-terminal fragments (CTX) ELISA quantification of soluble collagen fragments in the tendon (**C**) (N = 5). Mann-Whitney U test was used for statistical testing (** *p* < 0.01).

**Table 1 ijms-21-02835-t001:** Sequence of the primers used in quantitative real-time PCR.

Target Gene	Primer	Sequence
Scleraxis (Scx)	Forward Reverse	5′-acacccagcccaaacagat-3′ 5′-tctgtcacggtctttgctca-3′
Collagen IA1 (Col1A1)	Forward Reverse	5′-atgttcagctttgtggacctc-3′ 5′-gcagctgacttcagggatgt-3′
Collagen IIIA1 (Col3A1)	Forward Reverse	5′-tcccctggaatctgtgaatc-3′ 5′-tgagtcgaattggggagaat-3′
Integrin α1 (ITGA1)	Forward Reverse	5′-gatggggacgtcaacattct-3′ 5′-tgtggttaagacgctaccaaag-3′
Integrin α10 (ITGA10)	Forward Reverse	5‘-gaatcaggccgcatcctac-3‘ 5‘-aagtatcggagggcctgtg-3‘
Integrin α11 (ITGA11)	Forward Reverse	5′-gcagacgtcctctttaccaga-3′ 5′-gagctgtttgccttgacctc-3′
Hypoxanthine guanine phosphoribosyl transferase (HPRT)	Forward Reverse	5′-tcctcctcagaccgctttt-3′ 5′-cctggttcatcatcgctaatc-3′

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
