# Peer review of "Increased Collagen Turnover Impairs Tendon Microstructure and Stability in Integrin α2β1-Deficient Mice"

_ijms, 2020, doi:10.3390/ijms21082835_

Round 1
Reviewer 1 Report
The MS reads well and is a careful study of tendons in Itga2-/- mice.
I have the following comments:
- Formatting and text:
-Title section 2.5 : “Biomechanica Integrin”.. please correct.
- Sections 2.7 and 2.8 are duplications of sections 2.3 and 2.4 , please correct and renumber following sections accordingly.
- Figure legend figure 4 “poly acryl amide” rewrite into one word; Reference to Figure 4A is lacking.REvise figure legend to Fig 4.
- References:
In general not totally up to date, biased citing of own work, some central references related to work are lacking.
- In the discussion about what collagen-binding integrins bind to in different tissues the authors should consider including two references supporting the idea that integrin-mediated collagen binding indeed occurs to the fibrillar form of collagen and not to other macromolecules.
Chow, WY Scientific reports 2018:8 813809
Zhu, J Scientific reports 2018:8 16646
- Ref. 11 should be replaced by newer publication from the same group ( Ref. 11 Popova et al 2007 replace with Zeltz et al JCS 2016).
- A previous publication has addressed the role of integrins in human tendon and should be discussed. This includes scleraxis dependency on TGF-b, a2 integrin expression under conditions of released tensile strain and a2 function in tendon in relation to other integrins, in particular other collagen-binding integrins. ( Bayer, ML PLOS One 2014).
- Comments to text.
Line 209 “ It showed at distinct band at 40 kD” – please rephrase this sentence to make it more understandable.
4. Comments on results
For the Western in Figure 4A, please provide picture of full blot and positive control ( purified enzyme) and negative control (preimmune serum and secondary antibody).
Author Response
Reviewer 1
Dear reviewer, thank you for reading our manuscript and providing these helpful additions which improve the overall quality. In the following I will comment on the modification of the manuscript we made according to your requests:
- Formatting and text:
-Title section 2.5 : “Biomechanica Integrin”.. please correct.
- Sections 2.7 and 2.8 are duplications of sections 2.3 and 2.4 , please correct and renumber following sections accordingly.
- Figure legend figure 4 “poly acryl amide” rewrite into one word; Reference to Figure 4A is lacking.REvise figure legend to Fig 4.
We have removed the typos and wrong letters.
Addressing the duplicated paragraphs: When moving the manuscript to the IMSF/MDPI formatting, several paragraphs were duplicated. Word has added fitting paragraph numbers automatically; thus, this escaped our final quality control. We apologize for this.
- References:
In general not totally up to date, biased citing of own work, some central references related to work are lacking.
- In the discussion about what collagen-binding integrins bind to in different tissues the authors should consider including two references supporting the idea that integrin-mediated collagen binding indeed occurs to the fibrillar form of collagen and not to other macromolecules.
Chow, WY Scientific reports 2018:8 813809
Zhu, J Scientific reports 2018:8 16646
We have added a paragraph in which we discuss the Paper by Woltersdorf et al. the own work you question with mentioning the papers from Chow, WY Scientific reports 2018 and Zhu, J Scientific reports 2018 as you suggested. It is clear that the first paper just addresses collagen type II and chondrocytes whereas the papers you have mentioned focus on collagen type I as this is the major form in the tendon. Unfortunately, these papers did not show any cell binding experiments but stay solely on the structural level.
- Ref. 11 should be replaced by newer publication from the same group ( Ref. 11 Popova et al 2007 replace with Zeltz et al JCS 2016).
Ref 11 has been updated
- A previous publication has addressed the role of integrins in human tendon and should be discussed. This includes scleraxis dependency on TGF-b, a2 integrin expression under conditions of released tensile strain and a2 function in tendon in relation to other integrins, in particular other collagen-binding integrins. ( Bayer, ML PLOS One 2014).
We have added a paragraph discussing the finding of Bayer et al, which emphazises a possible role of integrin α2β1 in the tendon environment. Thank you for the suggestion!
- Comments to text.
Line 209 “ It showed at distinct band at 40 kD” – please rephrase this sentence to make it more understandable.
As it was not clear and properly referenced, we had already put the picture of the whole lysloxidase Western Blot into the supplemental data (Figure S3, now S4). When talking about the molecular weight we now have referenced the whole gel picture in the supplemental data. In addition, for the quantification, which is more imported from the point of scientific message, we refer to image of the isolated protein detection bands.
- Comments on results
For the Western in Figure 4A, please provide picture of full blot and positive control ( purified enzyme) and negative control (preimmune serum and secondary antibody).
As mentioned above, picture of the full blot was already provided in image S3 (now S4). The cross-reference was absent. We have redone the Western Blot under the same conditions accordantly to the suggestions using recombinant lysloxidase as positive control and pre-immune serum as negative control to validate the antibody. We have redone the western blot for 3 Integrin and 3 wild type samples as you see in the attachment. It is of notice that we get one specific band in the tendon samples, which is higher than the control. This may be due to glycosylation (the recombinant form obtained from CEDARLANE Burlington, Canada was from bacteria) or that we mostly detect the proform of lysyl oxidase. The negative control shows an unspecific signal for the positive control but no signal for the samples. For representation we have kept the original image since this is the sample, we have done the quantification using the fresh samples.

Reviewer 2 Report
General comments
The authors claimed that integrin alpha2beta1 deficiency leads to increased collagen turnover and impaired tendon microstructure and stability. However, the present data cannot exclude the possibility that other alpha family members, particularly, alpha1, can form a heterodimer with beta1, to compensate alpha2 deficiency and that compensated expression of alpha1beta1 can account for the observed phenotypes. The authors should conduct additional analysis to exclude this possibility.
Specific comments
#1. Section 2.5, 2.6, and 2.8, are repetition of section 2.1, 2.2, and 2.4, respectively. The repetition raised a serious question on the validity of the present observations as a whole.
#2. Figure 1. The authors should explain more clearly why the changed phenotypes under in vitro culture conditions cannot translate to the phenotypes of tendon in vivo.
#3. Figure 3D. Mann-Whitney U test cannot be used to analyze the data and a more appropriate method should be used for the analysis.
#4. Figure 4A and 4B. The authors should describe how to normalize the data.
#5. Figure 5A. The authors should describe how to normalize the data.
#6. Figure 5B. The figures are poor quality without showing the sizes of molecular markers and legends.
Author Response
Stange et al. bone 2013) therefore, we didn’t think the situation in tendon would be changed. However, since your remark is valid, accordingly we have done an expression analysis of the other integrin. It is described in paragraph 2.1 and the image is found now at figure S1 of the supplemental data. There was no compensation of the other collagen binding integrins found which is similar to bone cells.
Specific comments
#1. Section 2.5, 2.6, and 2.8, are repetition of section 2.1, 2.2, and 2.4, respectively. The repetition raised a serious question on the validity of the present observations as a whole.
Addressing the duplicated paragraphs: When moving the manuscript to the IMSF/MDPI formatting, several paragraphs were duplicated. Word has added fitting paragraph numbers automatically; thus, this escaped our final quality control. We apologize for this.
#2. Figure 1. The authors should explain more clearly why the changed phenotypes under in vitro culture conditions cannot translate to the phenotypes of tendon in vivo.
Thank you for this valuable comment, we specifically address this issue in lines 291-293 now, where we point out that due to the missing mechanical stimulation the overexpression of collagen degrading enzymes is not present in the tenocyte culture situation.
#3. Figure 3D. Mann-Whitney U test cannot be used to analyze the data and a more appropriate method should be used for the analysis.
As written in paragraph 4.13 Statistical analysis, Mann-Whitney U test was not used for the dynamic biomechanical testing but the two-way ANOVA. Since that was not obvious enough, we added this information to the caption of figure 3 as well.
#4. Figure 4A and 4B. The authors should describe how to normalize the data.
Description of the normalization has been added to paragraph 4.10 Lysyl oxidase determination.
#5. Figure 5A. The authors should describe how to normalize the data.
There was no further normalization; the Li-Cor Image Studio software quantified the signal in relation to the background calculation. The given data were the absolute numbers.
#6. Figure 5B. The figures are poor quality without showing the sizes of molecular markers and legends.
New image has been taken; molecular marker weights have been added
Round 2
Reviewer 1 Report
I have a minor comment applying to table 4 and Figure S1: gene names for integrins are denoted ITG, capital letters for human and Itg for mouse; hence I woudl suggest using Itga1, Itga10 and Itga11 instead of ITA1, ITA10 and ITA11.
Ref. 11 should be replaced (updated) as pointed out in previous review, in my revised version reference 11 is still the 10 year outdated references and has yet not been updated to 2017 reference.
Author Response
Dear Reviewer,
Thank you for the very fast answer. We have modified the manuscript according to your suggestions as follows:
I have a minor comment applying to table 4 and Figure S1: gene names for integrins are denoted ITG, capital letters for human and Itg for mouse; hence I woudl suggest using Itga1, Itga10 and Itga11 instead of ITA1, ITA10 and ITA11.
You are right. The analyzed samples are the genes and thus we have modified the nominator for integrins in the in the material an method part as well as in figure S1 according to your suggestion
Ref. 11 should be replaced (updated) as pointed out in previous review, in my revised version reference 11 is still the 10 year outdated references and has yet not been updated to 2017 reference.
Because of Corona caused switch to home office, the versions in which the reference was already altered got lost. I have updated the reference an additional time. This was not intentional.
Reviewer 2 Report
The authors modified the manuscript fully in response to the comments.
Author Response
Thank you for the revision.